# Vaccination-related attitudes and behavior across birth cohorts: Evidence from Germany

Claudia Diehl [1][�*], Christian Hunkler [2]☺

1 Cluster of Excellence: The Politics of Inequality, University of Konstanz, Konstanz, Germany, 2 Berlin Institute for Integration and Migration Research (BIM), Humboldt University Berlin, Berlin, Germany

☺ These authors contributed equally to this work.
* claudia.diehl@uni-konstanz.de

**Data Availability Statement:** The KiGGS data cannot be shared publicly because participants were assured that the data will only be used for scientific research and hence informed consent did only include restricted sharing of the data. The data

## Abstract

We use German KiGGS data to add to existing knowledge about trends in vaccination-related attitudes and behavior. Looking at vaccinations against measles, we assess whether a low confidence in vaccination and vaccination complacency is particularly prevalent among parents whose children were born somewhat recently, as compared to parents whose children belong to earlier birth cohorts. We further analyze how these attitudes relate to vaccination rates in the corresponding birth cohorts, and which sociodemographic sub-groups are more likely to have vaccination-hesitant attitudes and to act upon them. Results show that the share of parents who report "deliberate" reasons against vaccination has decreased across birth cohorts; at the same time, the children of these parents have become less likely to be vaccinated. This suggests that vaccination-hesitant parents became more willing to act upon their beliefs towards the turn of the millennium. Regarding efforts to convince parents and the public about the benefits of vaccination, the number of parents who think that vaccinations have serious side effects, or that it is better for a child to live through a disease, may have become smaller—but these parents are more determined to follow their convictions. Interestingly, the trend we describe started before the Internet became a widespread source of health-related information.

## Introduction

By the end of 2020, many countries had started vaccination programs against the SARS-CoV-2 virus, which causes Covid-19. Related to this, there has been a surge in public and academic interest in the phenomenon of vaccine hesitancy. After all, accepting vaccination is the key to ending—or not ending—the pandemic [1]. The term "vaccine hesitancy" has been used to describe a "delay in acceptance or refusal of vaccines despite availability of vaccine services" [2] and also attitudes that "doubt the benefits of vaccines, worry over their safety and question the need for them" [3, 4].

The topic is interesting from a social-science perspective because these vaccination-hesitant attitudes seem to at least partly reflect a mistrust of the actors and institutions involved in the development and distribution of vaccines, including not only "big pharma" and medical

("KiGGS_0100_v2.dta") are available to researchers from the Research Data Center of the Robert Koch Institute (contact via fdz@rki.de, +49 (0)30 18754 3200, https://www.rki.de/rdc) for researchers who meet the criteria for access to confidential data.

**Funding:** The author(s) received no specific funding for this work.

**Competing interests:** The authors have declared that no competing interests exist.

experts but also the state [3]. In this respect, it may be symptomatic of a broader political polarization in society [5]. In Germany, opposition to vaccinations against Covid-19 and to state-imposed measures to slow the spread of the Coronavirus plays an important role in the so-called "Querdenken" movement that gained momentum in 2020. Almost a year after the first vaccine was available in the country, vaccination rates against Covid-19 have remained comparatively low in Germany, especially among the population below the age of 60.

It is unclear whether hesitancy or refusal to get vaccinated against Covid-19 are a recent manifestation of increasingly skeptical public attitudes about vaccination. Empirical evidence for intuitively appealing statements regarding trends is often limited; this also applies to vaccination-hesitant attitudes. Using evidence from general population surveys and reports of medical professionals in Europe, some studies conclude that "(t)here is preliminary evidence that the prevalence of hesitancy and the challenges of addressing it are increasing over time" [3]. But longitudinal data on a longer time period that is based on "representative" and large enough population samples is scarce, if available at all, and vaccination-related attitudes cannot be inferred from vaccination rates. Not all those who doubt the efficiency and fear side effects of vaccinations refuse to have themselves or their children vaccinated, just like not vaccinating one's children may not always be the result of a deliberate decision (see next section for a more detailed discussion).

Against this backdrop, our paper seeks to make a modest yet substantive contribution to the existing knowledge about trends in vaccination-related attitudes and behavior—and the link therein. Using data from the German KiGGS survey [6], we describe first, whether and in which direction vaccination behavior has changed across cohorts and second, whether a low confidence in vaccination and/or vaccination complacency is particularly prevalent among parents whose children belong to more recent birth cohorts. Based on this, we explore, third, changes in vaccination behavior separately for parents who did or who did not report any reasons against vaccinations in the KiGGS survey. This is important because an overall trend of increasing vaccination rates may conceal the possibility that the subgroup of parents who are hesitant about vaccinations may have become more likely to have unvaccinated children. While previous analyses used KiGGS data to describe the vaccination rates of children whose parents have reasons against vaccination [7], those studies did not examine the changes across birth cohorts.

In brief, we show that the share of parents who report reasons against vaccination has decreased across birth cohorts rather than increased, the same is true for the share of children who are unvaccinated. But within the group of parents with deliberate reasons against vaccination, children have become less likely to be vaccinated across birth cohorts. We discuss three tentative explanations for this novel finding, and point out that much of the apparent trend happened before the Internet became a widespread source of health-related information.

## The 3C model and existing findings on changes in vaccine hesitancy

When it comes to explaining vaccination-related behavior, the "3C model" developed by the World Health Organization's Strategic Advisory Group of Experts on Immunization (SAGE) offers a theoretical framework for our descriptive and exploratory analyses. According to this model, vaccination behavior reflects *confidence*, *complacency*, and *convenience*. *Confidence* is "defined as trust in (i) the effectiveness and safety of vaccines, (ii) the system that delivers them . . . and (iii) the motivations of the policy-makers who decide on the need of vaccines" [8]. Critical events such as the publication of the infamous—and later retracted—article in *The Lancet* that wrongly linked vaccinations with autism may negatively affect this confidence [9].

*Complacency* is related to a low perceived risk of the particular disease targeted by a vaccine. Complacency should be higher for the polio vaccine than the measles vaccine, for example, since polio has been eradicated in Europe, so the likelihood of catching it is much lower than it is for measles, where outbreaks still occur. *Convenience*, or, as some authors prefer, *constraints*, relates to (but is not limited to) practical barriers to vaccination (such as language barriers when talking to a doctor, or a long distance to a vaccination facility). Betsch et al. have extended this model by a fourth factor, *calculation*, which "refers to an individual's engagement in extensive information searching" [8, 10]. Empirically, calculation is related to risk aversion and the "high availability of anti-vaccination sources, for instance, the internet" [8]. *Collective responsibility*, the perceived social benefits of a vaccination—that is, the reward of contributing to herd immunity and thus protecting others—was added as a fifth factor [8, 10].

One implication of this model is that vaccination-hesitant behavior does not always reflect a low confidence in vaccination, such as worries about side effects, or vaccination complacency, such as the belief that it is better for children to live through a disease. Some parents may just miss vaccinating their child for convenience-related reasons. Others may actually lack confidence in vaccinations or not find them necessary, but will still follow the protocols and vaccinate their child. Available evidence suggests, however, that individuals with reasons against vaccination have lower intentions to actually get vaccinated [10] and that the vaccination rates for their children are lower [7]. A low confidence in vaccination or vaccination complacency can also mean that parents are generally willing to vaccinate their children but hesitate to adhere to the protocol about when to apply a certain vaccination.

As mentioned in the introduction, available evidence on changes in vaccination-related attitudes over time is limited. Retrospective analyses of survey data from 149 countries suggest that vaccine hesitancy decreased in some countries and increased in others (including some in the European Union), but the period studied was only four years (2015–2019) [11]. Compared to the pre-pandemic European average, Germany was one country where public opinion was slightly more supportive of vaccination, according to the results of a Eurobarometer survey [12]. A recent report by the EU suggests that Germans' confidence in vaccination remained stable between 2015 and 2018 [13]. Studies based on biannual cross-sectional telephone surveys conducted by the German Federal Centre for Health Education between 2012 and 2018 document an increase in the proportion of Germans with (somewhat) favorable attitude towards vaccination, except for a drop in 2018 among those with lower education [14]. Vaccination *rates* in Germany have been slightly increasing rather than decreasing across birth cohorts [7, 15] and fluctuate at a high level in recent rates [16] but again, this does not necessarily imply that vaccine-hesitancy has decreased.

Media reports and anecdotal evidence suggest that for a potentially growing share of parents, vaccinating one's children is seen as a challenging decision that requires a careful weighing of costs and benefits, rather than a social script that is followed more or less automatically. The award-winning but heavily criticized 2017 documentary *Eingeimpft* is a prime example of this [17]. While opposition to vaccination is as old as vaccinations themselves [18, 19], it may be increasingly considered appropriate—or even obligatory—for parents to search out additional information about vaccinations rather than just following the protocols. At the same time, those who search online for information about vaccinations are likely to find more information that is critical about them [19, 20]. New information-seeking strategies on social media and the Internet's "medically subversive potential" [21] may have increased vaccine hesitancy.

For more recent birth-cohorts, many vaccination-critical websites are available, such as those on Facebook, that are highly—and supposedly more strongly than pro-vaccination sites —entangled with the sites of undecided users [22]. Via confirmation-bias, exposure to

vaccination-critical information may solidify the suspicions of individuals that vaccinations are unsafe, inefficient, or unnecessary. At the same time, acting on this belief may be encouraged within these networks of alternative "experts" whose opinions and recommendations differ from those of the "mainstream media," government agencies dealing with vaccinations, and medical experts. Research suggests that even short exposure to vaccination-critical websites increases risk perceptions of vaccination and intentions to not get vaccinated [20]. Vaccine misinformation on the Internet has been shown to reduce vaccination rates more strongly in areas with higher broadband coverage [23]. Against this backdrop, we will describe the vaccination rates of children of the parents surveyed in the KiGGS project and assess whether or not the share of parents with deliberate reasons against vaccination has increased across birth cohorts. Based on this, we then explore trends in vaccination behavior separately for parents with and those without reasons against vaccinations.

## Data and methods

The data requirements for analyzing vaccination-related attitudes and their link to behavior are high. As outlined, low confidence in vaccination, vaccination complacency, and changes therein over time cannot be inferred from vaccination rates, but only from analyses of self-reported reasons against vaccination. Analyzing survey data is challenging because the share of individuals with deliberate reasons against vaccination is small. Consequently, only surveys with large case numbers allow for systematic analyses of this issue. In addition, these surveys must contain valid information on vaccination behavior; this is essential when studying the link between vaccination-related attitudes and behavior. The longitudinal data necessary to analyze changes over time are currently unavailable. Studying vaccination-related attitudes and behavior across birth cohorts using the KiGGS survey is an alternative strategy even though it remains at best an approximation of what happens over time. When describing changes in vaccination rates, we supplement this with more recent data collected in the school entry examinations.

### The KiGGS survey

**Data.** In this article we used a customized version of the data collected in the baseline survey of the "German Health Interview and Examination Survey for Children and Adolescents" (hereafter KiGGS) study [6] that the Robert Koch Institute provided for the authors. The interviews as well as medical exams on children were conducted between May 2003 and May 2006. 87 percent of the respondents were mothers. Though not new, this study provides one of the few datasets that contain information on both vaccination-related attitudes and behavior from a broad range of birth cohorts. The KiGGS baseline study was approved by the ethics committee of Charité—Universitätsmedizin Berlin and by the Federal Commissioner for Data Protection. Participation was voluntary. Informed written consent was obtained from all parents and from adolescents aged 14 years and older. The data used for this paper ("KiGGS_0100_v2") are available to researchers via the Research Data Center of the Robert Koch Institute (contact via fdz@rki.de).

**KiGGS analysis sample.** The data include 17,640 observations for children and youth between the age of two months and 18 years. Since we are interested in who got their children vaccinated and who did not, we excluded children below the age of 24 months–which is the recommended age by which the vaccinations we are interested in should have been received. This leaves 15,780 observations on children born between 1985 and 2004. Additionally, we excluded children without a complete vaccination card (documenting the vaccinations), which reduces the analysis sample to N = 14,652. Furthermore, we limited our analyses to the birth

cohorts 1987 until 2002 because only those cohorts allow a robust description of vaccination rates based on more than 500 observations per birth year. That leaves N = 14,007 observations for analysis.

**Vaccination behavior.** Our behavioral dependent variable is the measles vaccination status at 24 months—the recommended age at which children should have received their first immunization. The measles vaccination is typically combined with the vaccination against mumps and rubella (MMR vaccine). In Germany, the first MMR vaccination should be completed when a child is between 11 and 14 months old (prior to 2001: 12 to 15 months), the second shot when the child is between 15 and 23 months old (prior to 2001: 5 to 6 years old [24, 25]). The measles vaccination status at 24 months was recorded in the medical exam part of the KiGGS study and was collected based on vaccination cards ("Impfausweise"). Analyses not shown here indicate that children who were not vaccinated against measles by the age of 24 months have on average received only 24 percent of all recommended vaccinations at the recommended age. This number is 80 percent for those who were vaccinated on time against measles.

**Vaccination attitudes.** In the survey part of the KiGGS study, *all* participants (including those whose children had received all recommended vaccinations) were asked whether they have reasons for not having their child vaccinated. We used this information to construct our second dependent variable, attitudes about vaccination (see Fig 1). We differentiated between three groups of parents. The first consists of parents who did not report any reasons against vaccination; the second group is made up of parents with "deliberate reasons" against vaccination who reported for example a fear of side effects (an indicator for "low confidence") or argued that it is better for a child to live though a disease (an indicator for "complacency"). The third group consists of parents who reported such reasons as being uninformed about vaccinations ("low convenience"). This group, which we call parents with "convenience-related

**Fig 1. Survey instrument for reasons against vaccination.** * See Table 1 for details on reasons reported in the open answer category, own translation of survey instrument (original question in German: Hatten Sie Gründe, Ihrem Kind Impfungen nicht geben zu lassen?).

**Table 1. Share of parents reporting reasons to not vaccinate by measles vaccination status of child at 24 months.** KIGGS data, shares weighted, obs. not weighted, N = 13,507 observations; a small minority of 82 parents gave deliberate as well as convenience-related reasons, from 757 parents who gave confidence reasons 475 also gave complacency reasons. * Preterm birth and egg white allergies are not considered contraindications to most vaccines.

| Reason not to vaccinate | 1st measles vaccination received in time | | | | Total | |
|---|---|---|---|---|---|---|
| | No | | Yes | | | |
| | % | obs. | % | obs. | % | obs. |
| Deliberate reasons: one or more deliberate reasons(s) | 17.6 | 662 | 3.7 | 320 | 7.6 | 982 |
| Confidence: one or more confidence reasons | 13.5 | 515 | 2.8 | 242 | 5.8 | 757 |
| Fear of side effects; side effects with previous vaccination; egg white protein allergy*; preterm birth* | 13.3 | 508 | 2.7 | 232 | 5.7 | 740 |
| Fear of vaccination procedure | 0.2 | 10 | 0.1 | 10 | 0.2 | 20 |
| Complacency: one or more confidence complacency reason(s) | 12.6 | 494 | 1.7 | 149 | 4.8 | 643 |
| Undergo illness; respective vaccination regarded unnecessary | 12.1 | 474 | 1.6 | 132 | 4.5 | 606 |
| Respective vaccination not yet regarded necessary, if need be later | 1.1 | 47 | 0.2 | 19 | 0.5 | 66 |
| Other deliberate | 1.9 | 64 | 0.6 | 52 | 0.9 | 116 |
| Uncertainty about risk/ benefit/ necessity of respective vaccination; lived through respective disease already, physician/ naturopath advised against it; dissuasion by other person (not a physician or naturopath); other reason (not coded) | | | | | | |
| Convenience-related reasons: one or more convenience-related reason(s) | 3.3 | 114 | 0.7 | 62 | 1.4 | 176 |
| Uninformed | 0.4 | 15 | 0.1 | 6 | 0.1 | 21 |
| Forgotten | 2.8 | 99 | 0.6 | 52 | 1.2 | 151 |
| Organizational reasons (access/ health care provisions) | 0.2 | 5 | 0.1 | 4 | 0.1 | 9 |

reasons," may include recent immigrants who are not yet familiar with the health system and/ or face language barriers when speaking with a doctor.

Other reasons given in the open answer section of the questionnaire were coded into the respective groups (see Table 1 below), such as that it was planned for later, or that a doctor or "naturopath" had told the parent it was not necessary (on the link between homeopathic practitioner and vaccination behavior and recommendations see [26, 27]). Note that the KiGGS questionnaire was constructed before the 3C model was developed. Therefore, the KiGGS items we used to operationalize confidence, complacency and convenience do not perfectly capture the theoretical dimensions and were not validated for that purpose. Moreover, information on the other antecedents of vaccination behavior discussed above, *calculation* and *collective responsibility*, is not available in KiGGS.

**Independent variables.** We examined the following variables that affect vaccination-related attitudes and behavior according to previous studies [26]. Levels of education—which are negatively related to confidence in vaccinations—were measured as the highest degree of schooling achieved by parents ("low" = lower or medium secondary school degree or less; "medium" = higher secondary school degree ("Abitur"), or vocational training degree; "high" = all forms of tertiary education). We included the age of the mother when the child was two years old, because according to previous research, older mothers (age 36 or more when the child is two) are more likely to not vaccinate their child [26]. Older siblings were also taken into account, because their presence in the household may affect both convenience and complacency. On the one hand, visits to the pediatrician may be more of a routine for parents who already have a child. On the other hand, parents of children with an older sibling are statistically less likely to plan another child in the future (given that few families in Germany have three or more children). Thus, there is a lower risk that the unvaccinated second child (that is, the child the KiGGS data refer to) could potentially pass the measles on to a newborn (and

necessarily unvaccinated) third child. This could increase complacency by reducing the perceived necessity to vaccinate a child that has one or more older siblings. We also took into account migration background (one or both parents or the child born abroad) and controlled for the size of the municipality. When analyzing vaccination behavior, we also looked into the sex of the child and whether and at what age it was first in non-parental supervision, e.g. a nursery or kindergarten. The latter could trigger considerations of collective responsibility. S1 Table provides an overview of all independent variables.

**Groups.**    We analyze three groups: First, the majority of parents (91.4%) who report no deliberate and no convenience reasons; second, parents who reported deliberate reasons (7.7%) and; third, parents who reported convenience reasons (1.5%). The latter two groups do partially overlap, a small minority of parents (0.6%, N = 82) reported deliberate *and* convenience reasons. We cannot analyze this small group separately, and therefore included them in both groups.

**Statistical analysis.**    For the multivariate analyses, we estimated logistic regressions on the binary coded vaccination behavior (0 = not vaccinated; 1 = vaccinated) and on the binary coded vaccination attitudes (0 = no reasons; 1 = one or more deliberate reason, or respectively 1 = one or more convenience reason). All reported regression models show very low multicollinearity. The variance inflation factor (VIF) is 1.15 at most (typically a value of VIF = 10 or greater is considered cause for concern). For the multivariate analyses, we used multiple imputation for the independent variables to maximize the use of the available information and to minimize the complete case analysis bias [28]. Independent variables were imputed 25 times using chained equations. In chained equations, missing values are iteratively replaced using a sequence of univariate imputation methods with fully conditional specifications of prediction equations. Our imputation system included all variables used in the multivariate analyses. For the cohort categories and the indicator on whether the mother was age 35 or younger when the child was two years old, we imputed the underlying metric variables (year of birth of the child and age in years of the mother). The proportion of missing data is small for most independent variables (see S1 Table). Still, listwise deletion would lead to using only 93.3% of the available observations. We used the weight variable supplied by the KiGGS Study Group in all analyses. The weight accounts for sampling design and non-response (basically the inverse of the sampling and response probability using region, age, sex, and education [29]). Robustness analyses without weighting and on non-imputed data revealed no substantial differences compared to the results reported. For the data preparation and analyses we used the statistical software package Stata 15.1 SE. The scripts to prepare the data and replicate the analyses are available via an Open Science Framework repository (https://doi.org/10.17605/OSF.IO/36RB8).

## The school entry examination data

The school exam data ("Schuleingangsuntersuchungen", we use data made available by the Federal Health Monitoring Information System [30]) allows us to capture more recent developments in vaccination rates. It is available for all federal states from 2005 onward [31]. The data include vaccination rates by federal state and year based on entry exams. The rates are based only on children whose vaccination card was available at the time of the exam (for example, 91.2 percent in 2005). Since children without vaccination cards are more likely to not be vaccinated, the following analyses may slightly overestimate vaccination rates [31].

## Results: Vaccination-related behavior and attitudes across cohorts

We start out by describing the share of children who received the first shot against measles on time, across time and birth cohorts. Based on this, we assess the share of parents who have

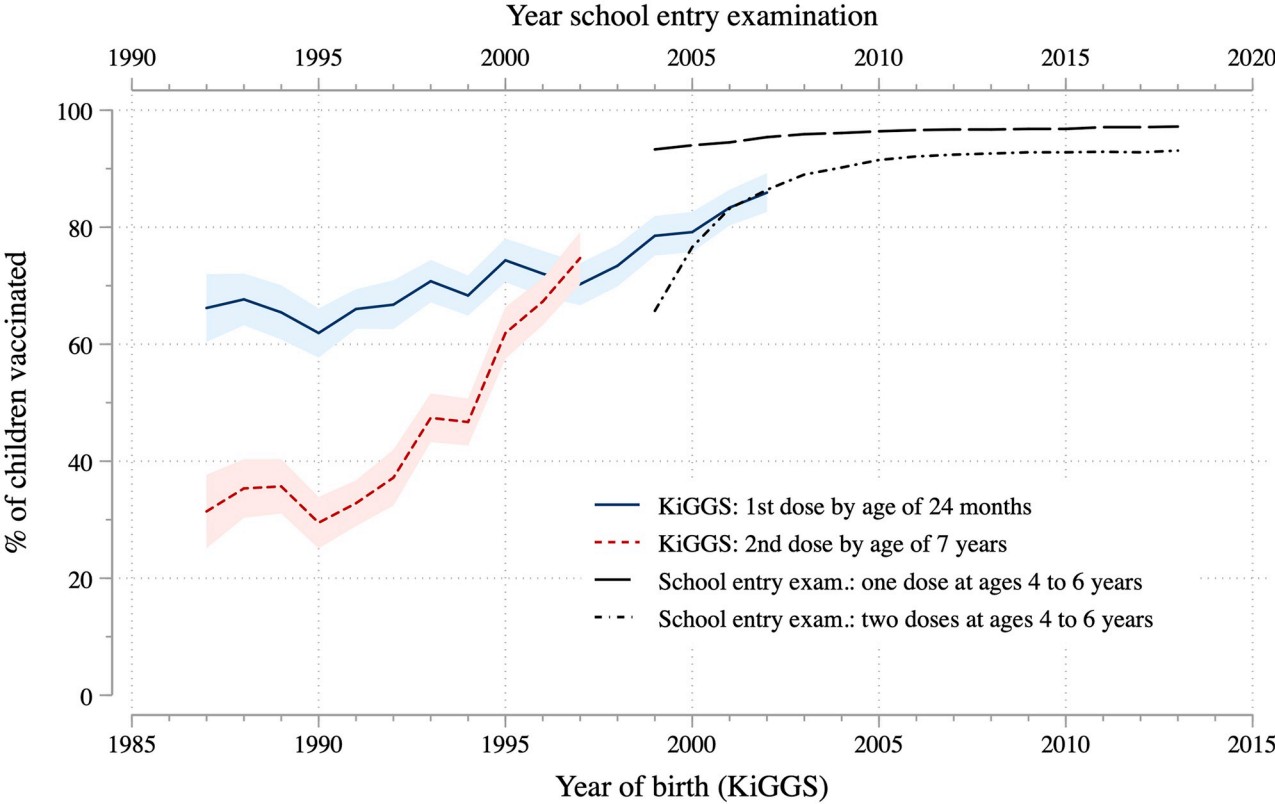

**Fig 2. Measles vaccination rates over time.** KiGGS data weighted, 14,007 observations for first measles dose by age 24 months (between 535 and 991 observations per year of birth) and 9,715 observations for second dose by age 7 years (between 535 and 975 observations per year of birth), point estimates and 95 percent confidence intervals. Note that in the school entry exam data in 2004 only, vaccination rates for two states (Saxony and Rhineland-Palatine) were not available.

deliberate or convenience-related reasons against vaccination in different birth cohorts, and describe the sociodemographic composition of both groups. Finally, we analyze the share of unvaccinated children and their parents' sociodemographic composition by types of vaccination-hesitant attitudes (deliberate reasons, convenience-related reasons, no reasons) across cohorts.

## Changes in vaccination rates

Fig 2 shows the measles vaccination rates for the birth cohorts of 1987 until 2002 (1997 for the second dose), based on the KiGGS study and data from school entry examinations. Whereas the KiGGS data record the vaccination status of children by their year of birth, the vaccination rates from the school entry examination refers to about five year old children. A child born in 2000 will thus appear in the school entry examination data approximately in 2005. To account for this, we shifted the upper horizontal axis five years to the left to match with the rates calculated from the KiGGS data that are displayed by year of birth.

Measles vaccination rates overall increased until 2010 and remained stable at a rather high level afterwards. The share of children who had received their first vaccination by the age of 24 months increased from about 65 percent to more than 80 percent between 1987 and 2002 (see lower axis). This trend is also visible in the school entry exam data. The share of children who

had received their first vaccination at a later age, namely between four and six years old, is more than 90 percent in 2004/05 and approaches 95 percent afterwards (see upper axis). This trend toward a higher share of vaccinated children is even stronger for the second vaccination, as suggested by the rising share of children who received it by age seven. This share increased by more than 40 percentage points between 1987 and 1997, and it continued to rise in the more recent past, according to school entry exam data. While the different age criteria applied in the KiGGS data and school entry exam data are not ideal when describing the measles vaccination rate over time, the overlap in the birth cohorts from 1999 to 2003 indicates that many children receive their first measles shot later than recommended: vaccination rates are higher when these children are observed four to six years later in the school entry examination. In sum, the share of children who have received their measles shots has been rising, but a substantial share—at least 7 percent of children in 2018 (based on school entry exam data)—have not been fully vaccinated against measles on time.

We conducted additional analyses separately for East and West Germany. These graphs show that, most likely related to the disruptions surrounding German reunification, vaccination rates for children born around 1990 in Eastern Germany dropped sharply from the very high levels of children born before reunification. Levels increase and converge with (rising) rates of children born in West Germany in later-born cohorts (see S2 Fig). Data from school-entry examinations show only small remaining differences between both regions (see S3 Fig).

## Parents with reasons against vaccination across birth cohorts

As described above, all parents (including those whose children had received all recommended vaccinations) were asked whether they had reasons for not having their child vaccinated. Of these, 9.7 percent said yes, 87.6 percent said no, and 1.7 percent said do not know. For 1.0 percent of respondents the answer is missing; 8.2 percent gave one or two reasons for not vaccinating their child, and only a few gave more. Table 1 displays the share of parents reporting reasons against vaccinations by the vaccination status of the child.

Among parents of unvaccinated children, 17.6 percent have deliberate reasons for not vaccinating, most importantly the fear of side effects (indicator for "low confidence") and the belief that it is better for the child to live through the disease (indicator for "complacency"). Obviously, not all parents who have deliberate reasons against vaccination act upon these beliefs when it comes to measles: 3.7 percent of parents of vaccinated children have deliberate reasons against vaccination. Note that the question about reasons against vaccination does not refer to a specific vaccination (we will discuss this in greater detail below). Fewer parents report convenience-related reasons, such as feeling uninformed or having forgotten to vaccinate their child. Overall, the measles vaccination rate of children of parents with deliberate and/or convenience-related reasons is around 35%, while the rate is at 74.7% for children of parents without any reason.

Fig 3 shows that the share of parents with deliberate reasons is highest (9.9 percent) among parents of children who were born in the 1991/92 birth cohort, and has slightly decreased to 5.9 percent among parents of the most recent birth cohort. The share of parents claiming that they forgot to have their children vaccinated or were uninformed about it is much smaller in comparison but follows a similar trajectory across cohorts. In East Germany, we do not observe such a clear pattern. See S4 Fig for reasons against vaccination across cohorts separately for East and West Germany.

In Fig 4 we take a closer look at the characteristics of parents who reported different reasons against vaccination as compared to those who did not report any reason. These models confirm the slight decrease in the share of parents with deliberate reasons against vaccination

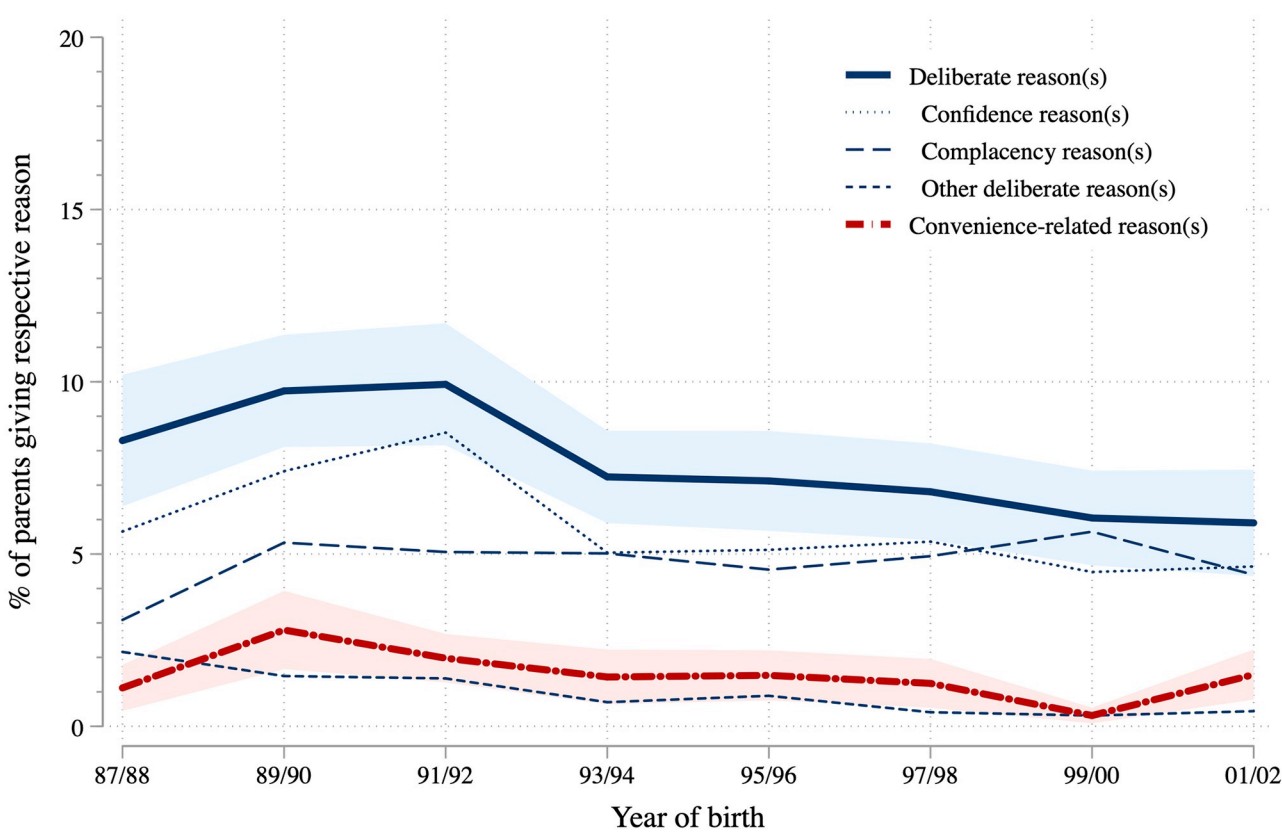

**Fig 3. Reasons to not vaccinate over time.** KiGGS data weighted, depending on reason between 13,517 and 13,784 observations (between 1,216 and 1,933 observations per two-year birth cohort), point estimates and 95 percent confidence intervals.

(blue) or with convenience-related reasons (red), as seen in Fig 3. Compared to the oldest cohort (born 1987–1990), the share of parents with deliberate reasons against vaccinations dropped by 3.9 percentage points in the youngest cohort (born 1998–2002). Migrants, respondents living in East Germany, and younger mothers are less likely to report deliberate reasons against vaccination, while parents with high levels of education and those living in large cities are more likely. In turn, parents who report convenience-related reasons against vaccination have lower levels of education and are more likely to have an older child. In sum, both groups differ substantively in their sociodemographic profile.

## Vaccination behavior of parents with and without reasons against vaccination across birth cohorts

Turning to the link between vaccination-related attitudes and behavior, we compare the vaccination status of children in different birth cohorts separately for those parents with deliberate reasons against vaccination, parents with convenience-related reasons, and parents without any reason against vaccination (Fig 5). This exercise yields a noteworthy finding: While vaccination rates increase across cohorts among those parents who report no reasons against vaccination, we see a declining compliance among parents with deliberate reasons against vaccination. Since the latter group is a small minority (about 6 to 10 percent, see Fig 3), this trend is perfectly compatible with the overall increasing vaccination rates described in previous studies and shown above in Fig 2. Due to small case numbers, the pattern for parents with

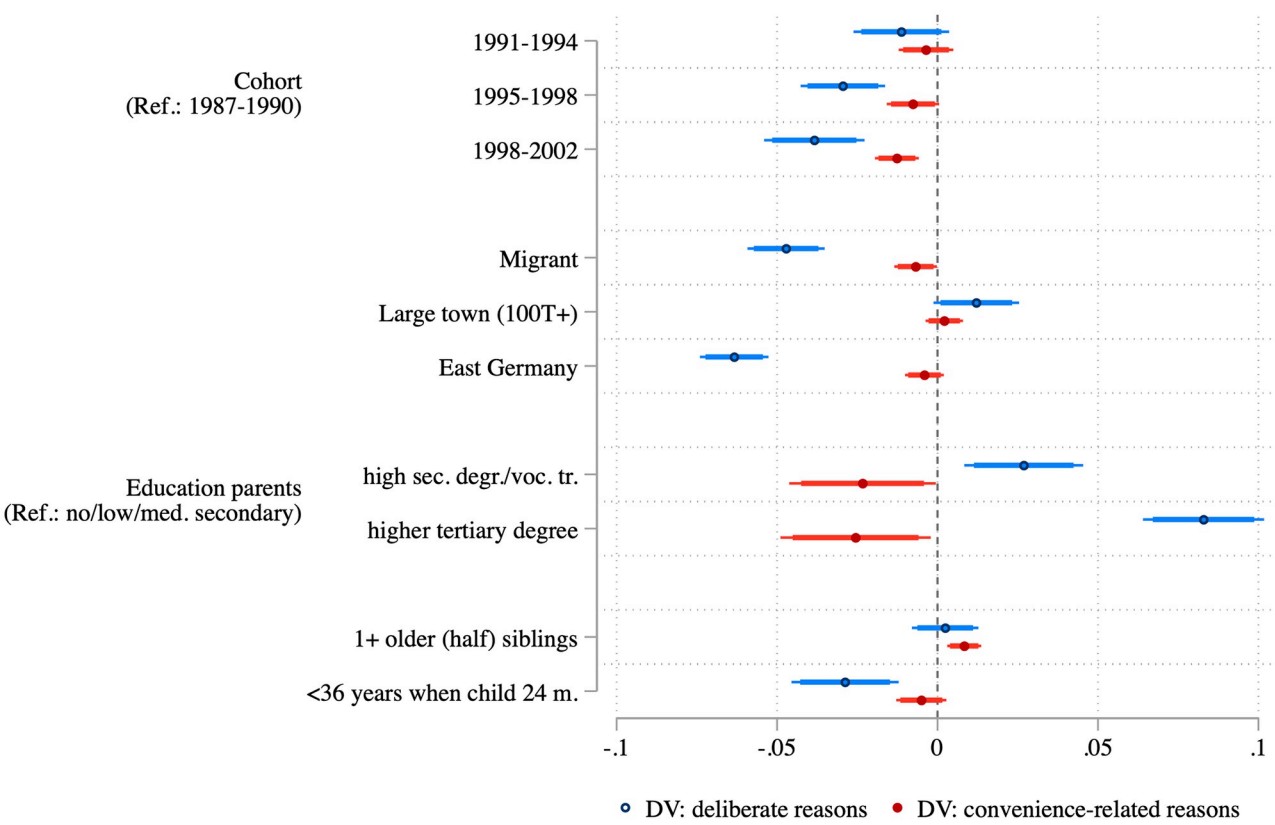

**Fig 4. Average marginal effects from logistic regressions predicting deliberate and convenience-related reasons to not vaccinate.** Point estimates, 95 percent confidence intervals (thin bars), and 90 percent confidence intervals (thick bars) showing average marginal effects based on multiply imputed weighted data (see S2 Table).

convenience-related reasons is less clear. See S5 Fig for separate analyses for East and West Germany that show overall a similar pattern for both regions.

We supplement these analyses by presenting average marginal effects (see Fig 6) from three regressions predicting the vaccination behavior for parents who report deliberate reasons against vaccination (blue), those who report at least one convenience-related reason (red), and those who do not report any reason (grey).

Results confirm that among those who do not have any reasons against vaccination—the largest group by far—children from younger cohorts are *more* likely to be vaccinated. For children whose parents have deliberate reasons against vaccination, we see again a different pattern across birth cohorts: those in the more recent cohorts (from 1995 on) are *less* likely to be vaccinated. Compared to living in a German-origin family, living in a migrant family reduces the chances of being vaccinated for children whose parents do not report any reasons against vaccination. Living in Eastern Germany has an opposite (that is, positive) effect in this group and in the group of those with convenience-related reasons. Having educated parents has a strong positive effect on being vaccinated in the group with no reasons or convenience-related reasons, but not in the group reporting deliberate reasons. As a reminder: Highly educated parents are more likely to report deliberate reasons against vaccination. An older sibling reduces the chances of being vaccinated in all three groups (only by tendency in the low-convenience group). As argued above, an older sibling may lower the perceived individual benefit

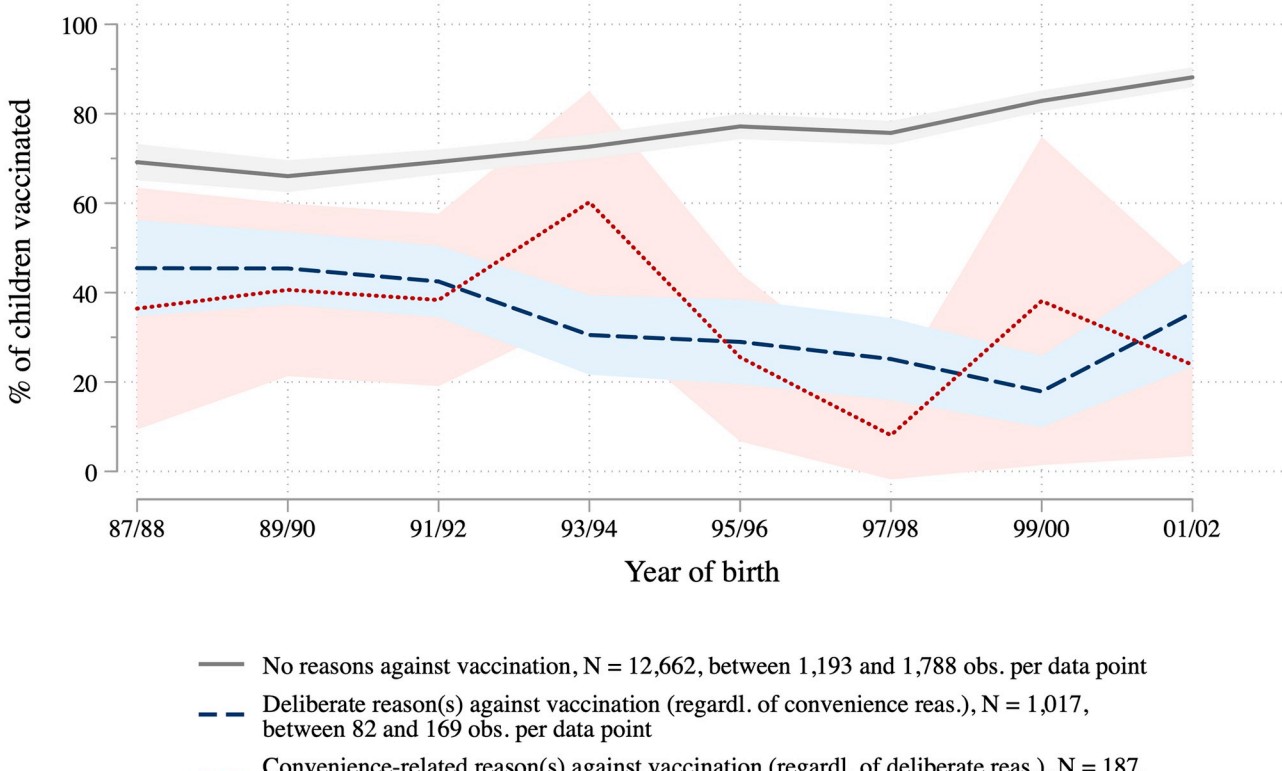

Fig 5. Measles vaccination behavior over time for parent groups defined by (not) having deliberate or convenience-related reasons against vaccination. KiGGS data weighted, point estimates and 95 percent confidence intervals.

of the measles vaccination, and this obviously has an impact, even though the collective benefit (supporting herd immunity) is, of course, the same.

### Predicted vaccination rates: How likely are children of "typical" parents to be vaccinated against measles?

Based on a regression model integrating the findings depicted in Fig 6, we finally calculate predicted vaccination rates for exemplary cases to present our findings more intuitively. Fig 7 contrasts "extreme" groups based on the variables that have proven important in explaining vaccination behavior, namely reporting deliberate reasons against vaccination, and level of education. It is based on a logistic regression of vaccination behavior for the whole sample. In addition to the independent variables used above, we added the binary indicators for deliberate and for convenience reasons. Moreover, we included four two-way interaction terms between the two reason variables and cohort as well as education (see S4 Table for details). The most important result is that deliberate reasons against vaccination increasingly make a difference when it comes to explaining whose child is vaccinated and whose is not, especially among more educated parents. In the oldest cohort, the likelihood of being vaccinated differs by 4 and 27 percent between children of parents with and without deliberate reasons against vaccination (lower and higher levels of education, respectively). But this gap has increased across cohorts, it is most pronounced in the youngest cohort, where it ranges from 45 to 61 percent

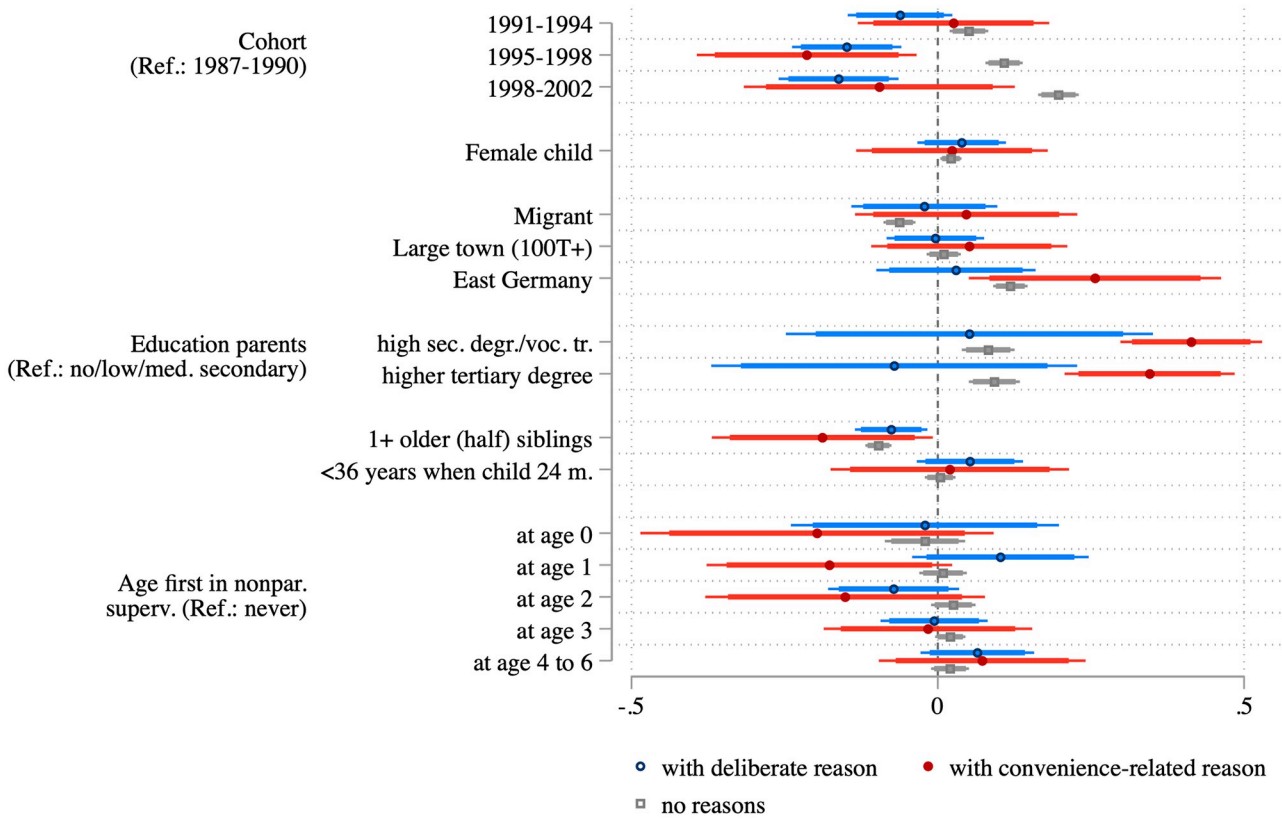

**Fig 6. Average marginal effects from three logistic regressions predicting vaccination of the child in different groups of parents.** Point estimates, 95 percent confidence intervals (thin bars), and 90 percent confidence intervals (thick bars) showing average marginal effects based on multiply imputed and weighted data (see S3 Table).

(lower and higher levels of education, respectively). In addition, even though confidence intervals partially overlap, education tends to have an opposite effect on the vaccination behavior for those with and those without deliberate reasons against vaccination: among parents who worry about side effects or think it is better for a child to live through a disease, the children of highly educated parents are less likely to be vaccinated. The opposite is the case among parents who do not report deliberate reasons against vaccination. In sum, highly educated parents with deliberate reasons against vaccination are particularly and—across age cohorts—increasingly unlikely to have their children vaccinated.

## Discussion and limitations

Our analysis of KiGGS data has shown that vaccination rates have increased and that the share of parents who report reasons against vaccination has decreased across birth cohorts. This may indicate that the efforts to convince parents and the general public about the benefits of vaccination have been successful—even during a time when vaccination-critical information has become easily accessible on the Internet [19]. Health education and an increasing health literacy may have played an important role in this respect. While long-term data on this is unavailable for Germany, existing studies show that health literacy comes along with higher trust in medical experts and a greater capability to identify reliable information [32]. Regarding the link between vaccination-related attitudes and behavior, results show that vaccination

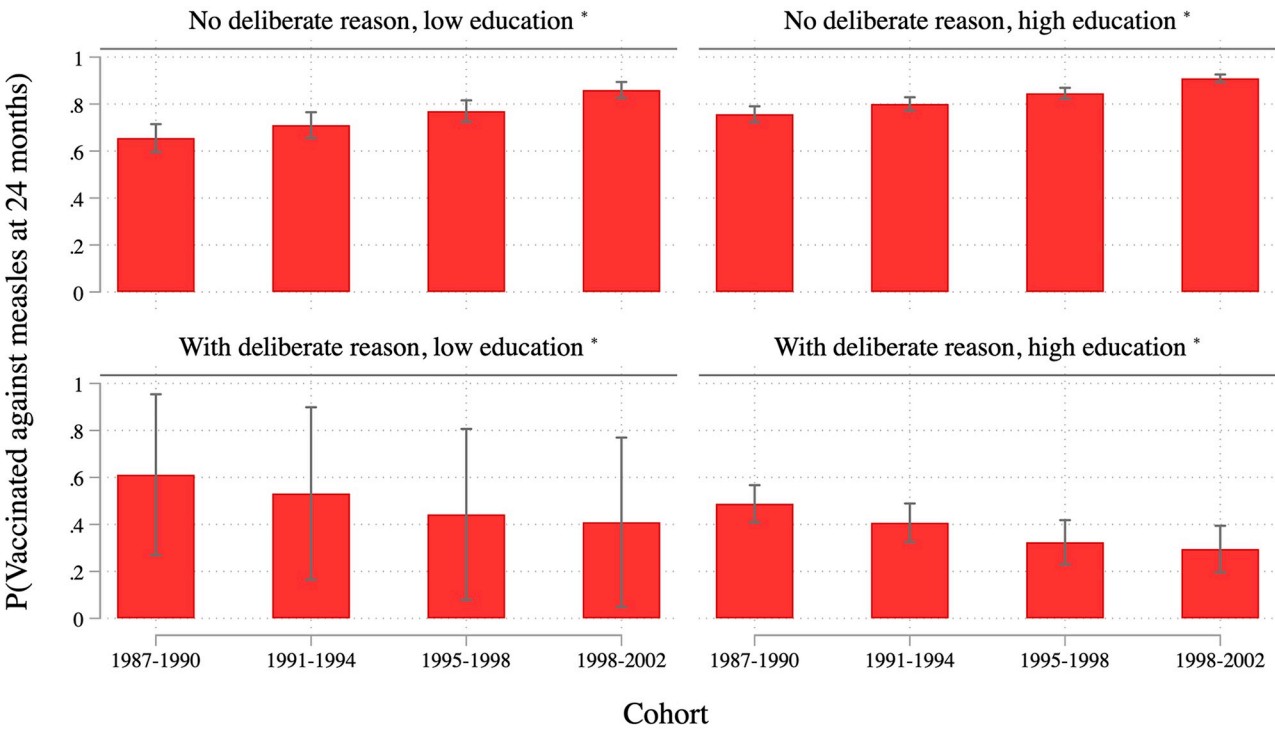

**Fig 7. Predictive margins from logistic regression predicting vaccination behavior.** Predictive margins and 95 percent confidence intervals based on logistic regression with multiply imputed and weighted data (see S4 Table). * non-migrant family with female child in large West-German town without older siblings, younger mother, going to nonparental supervision at age 3, and without non-deliberate reasons.

rates declined across birth cohorts in the group of parents with deliberate reasons against vaccination (low confidence and/or complacency). Put differently, in this group, younger children are less likely to be vaccinated by age 24 months than older children are. We can think of three possible explanations for these findings.

First of all, parents whose children were older at the time of the interview (the children of the oldest birth cohort—born in 1987/88—were close to adulthood when the KiGGS interviews were conducted between 2003 and 2006) could have become hesitant or skeptical about vaccination only recently, or at least after their children were vaccinated. This is why their compliance was higher when their children were young. While we cannot fully exclude this interpretation based on the cross-sectional data at hand, we believe that several arguments speak against it. Above all, it seems likely that parents make up their minds about vaccination at a time when the salience of the issue is high—that is, when their children are young and they must decide whether or not to have them vaccinated. To be sure, one can become skeptical about vaccination later in life. However, little societal debate, not to mention pressure, evolves around vaccinations that are applied later in life such as the one against FSME (tick-borne encephalitis) or vaccinations for travelers, e.g. against meningitis. The topic should therefore be much less salient once the decision about children's vaccinations has been made. Moreover, theories of cognitive dissonance also suggest that a negative feeling is evoked when a person behaves in a way that contradicts his or her beliefs without an external justification for doing so. As a consequence, people search selectively for information that confirms a decision that was previously made [33]. One could argue that critical events such as the publication of the above-mentioned study in The Lancet linking the measles vaccination to autism make changes

in post-decisional attitudes more likely, especially when quite some time has passed. But again, these discussions should be more relevant for parents of young children than for the general population. In addition, the increase of unvaccinated children among parents who are skeptical or hesitant about vaccination occurs rather steadily across birth cohorts. This, too, speaks against such "period effects."

If the idea that parents have only become more skeptical about vaccination after their children were vaccinated is discarded for these reasons, a second interpretation becomes more likely. It is possible that among these parents, those with younger children—who, as a reminder, are less likely to report deliberate reasons against vaccination than are parents of older children—were more willing to act upon their belief after the turn of the millennium than they were in the early 1990s. Based on our calculations, back then over 45 percent would still follow their pediatrician's advice and the official recommendations about vaccinations, but this share has decreased to about a quarter in the youngest cohort (compare Fig 5). The abundance of vaccination-critical information to be found on the Internet, along with the alternative "experts" who confirm and reinforce these doubts, may have weakened the role of medical experts who argue in favor of vaccination among those parents who were skeptical or hesitant to begin with.

This interpretation may sound convincing, but data on Internet coverage and social media use shows that the drop in the share of skeptical parents whose children are vaccinated occurred mostly before the late 1990s. Until this time, Internet coverage was still quite low in Germany; it only reached a substantial level of 30 percent toward the turn of the century (see Fig 8). But that was about when the trend of declining measles vaccination rates in complacent

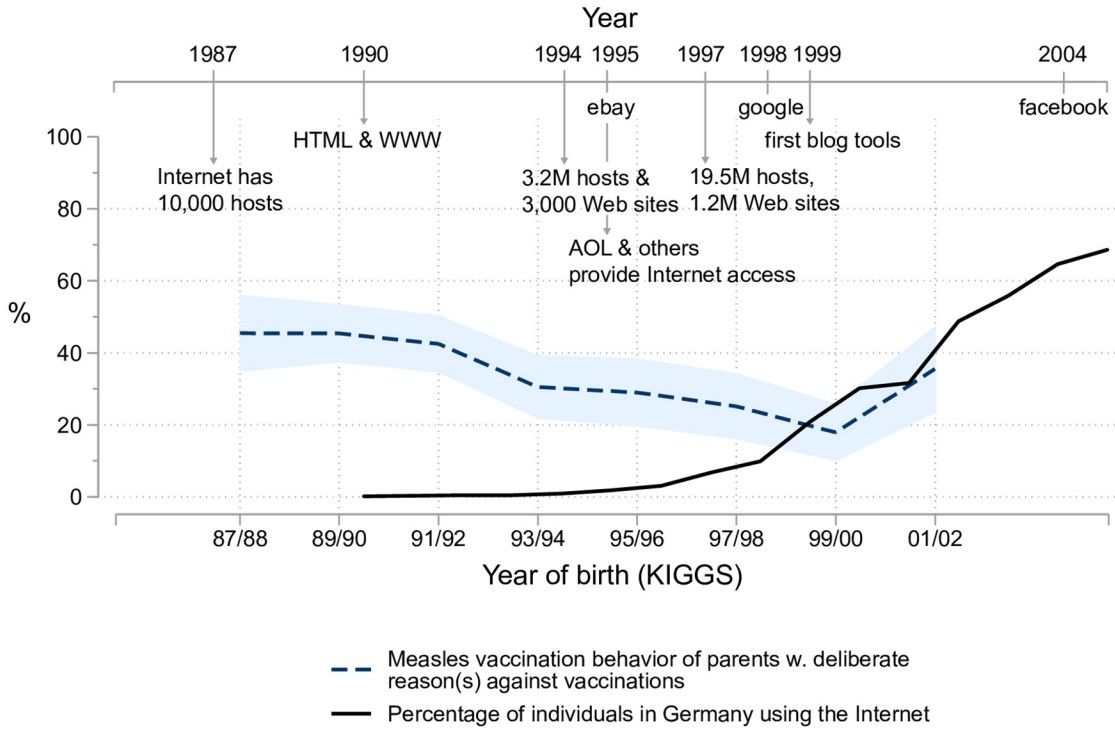

**Fig 8. Development and use of the Internet and measles vaccination rates for children of parents with deliberate reasons against vaccination.** KiGGS data weighted, N = 1,210. Internet usage data from International Telecommunication Union (ITU)—the United Nations agency for information and communication technologies. Internet milestones: [36, 37] and company websites.

or low-confidence families apparently stops. Rates for those born in 2001/02 are actually higher than in the previous cohort. Obviously, processes such as a "dethroning" of medical experts started earlier and were particularly appealing to more educated parents. Influential books, like Ivan Illich's *Disabling Professions* and *Medical Nemesis*, published in the 1970s [34, 35] which tackle the "sickening power" of the "so-called health professions" may have played a role in this process.

A third, more prosaic explanation suggests that our finding is the result of a success story. Previous studies document rising vaccination rates across birth cohorts in Germany, and our findings point towards a declining share of parents with complacency/low confidence in vaccination. This may reflect efforts to convince parents and the public about the benefits of vaccination. As a consequence, the remaining share of those parents who think it is better for a child to live through a disease, or that vaccinations have serious side effects, could have become more selective and more "radical" in terms of their beliefs—and thus more determined to act upon them.

In sum, the interpretations of our rather straightforward results must remain tentative and our study faces several limitations. Most importantly, our analyses are not based on—currently unavailable—longitudinal data and thus remain an approximation of how attitudes influence vaccination behavior over time. Related to this, the attitudes we analyze were captured retrospectively between 2003–2006. The limited accuracy of recollections, a so-called "recall bias" in the information retrieved during a survey, is a general problem in retrospectively collected data, especially when it comes to attitudinal data. Another limitation refers to the measurement of vaccine hesitancy in the KiGGS survey which reflects the limited state of research on that concept at that time. In addition, the time period we are able to focus on ends in the middle of the first decade of this century. We thus cannot say anything about how vaccine hesitancy has changed between then and now. We also want to emphasize that while we have shown that parents who did not report any reasons against vaccination mostly have vaccinated children, a small share of them—but a numerically rather large group—has unvaccinated children. With the data at hand it remains unknown whether and why they decided against vaccinating their children or whether they faced practical barriers in doing so or forgot about it. A final limitation to our analyses is that we do not know whether deliberate reasons against vaccination were related to a particular vaccine. In the early 1990s, for example, there was debate about the side effects of the vaccine against pertussis. One might object that parents with a low confidence in this specific vaccine reported deliberate reasons against vaccination but still vaccinated their children against measles. This would be an alternative explanation of our finding about a decline in compliance of parents with reasons against vaccination. Robustness checks reveal, however, that this is not the case (see S1 Fig). We observe the same pattern for *all* vaccinations recommended in Germany and not just for the measles vaccination for the whole observed period. The vaccination rates for children of parents with deliberate reasons decline, while the rates increase when parents reported no reasons. In addition, we see the same pattern when we look at complacent parents and those with a low confidence in vaccination separately, and again this holds not only for the measles vaccination but for all recommended vaccinations (see S1 Fig).

Despite the shortcomings of the data we used in our analyses, we thus believe that we have contributed to the knowledge on trends in vaccine hesitancy and its behavioral repercussions. So far, there is no alternative approach for those interested in catching at least a glimpse of the dynamics of vaccination-related attitudes and behavior over a long period of time.

## Conclusion: Lessons for today

With vaccinations underway against Covid-19, those who are opposed to vaccination are quite vocal. The German "Querdenken" movement provides a good example of the link between

vaccine hesitancy and a broader political polarization. But the recent interest in vaccine hesitancy seems at odds with today's high vaccination rates. Against this backdrop, our motivation for writing this paper was twofold. On the one hand, we wanted to consider this issue by analyzing a dataset that contains valid information on changes across cohorts in parents' confidence in vaccination, their vaccination complacency, and other reasons that may keep them from vaccinating their children. On the other hand, we wanted to analyze vaccination behavior—including that of the small subgroup of parents with deliberate reasons against vaccination—in greater detail. The data we used are neither longitudinal nor particularly new, but they allow for cohort-specific analyses of the behavior and attitudes we are interested in.

We identified a group of roughly five percent of parents with vaccination complacency/ low confidence in vaccination who act on their beliefs. Existing studies show that the small but determined group of those who outright oppose the vaccination program, are hard to reach with strategies such as increasing the motivation and removing practical barriers [10]. If vaccination-related attitudes and behavior were indeed considered a symptom of a broader political polarization, as outlined in this paper's introduction, they might more appropriately be described as "radicalization at the margins" rather than "polarization." To be sure, our study is not about vaccination against Covid-19 and uses data that was collected more than a decade ago. While we could not take into account distrust in institutions as a factor predicting vaccine hesitancy in our analysis, the link between a lack of confidence in vaccines and in the institutions that support vaccine development and organize its distribution, notably the government, is empirically well-established by now [38, 39]. In the light of the current debate and given the complete absence of long-term data on vaccine hesitancy, we believe that one important contribution of our study is to show that the opposition to vaccination in Germany is not the culmination of a long-term trend towards more vaccine hesitancy. For those interested in the topic of political polarization, it may seem good news that the group who strongly opposes vaccinations seems rather small, according to our study and more recent research. From an epidemiological perspective, however, it is still too large when it comes to tackling very infectious diseases such as measles or, for that matter, new mutations of the Coronavirus.

## Supporting information

**S1 Table. Descriptive statistics.** Non-imputed and (last column) multiply imputed KIGGS data.
(PDF)

**S2 Table. Regression estimates of deliberate and convenience reasons to not vaccinate depicted in Fig 3.** Odds-Ratios from logistic regressions using weighted and multiply imputed data; McFadden's R2 derived using Rubin's combination rules and ignoring the clustered data structure; Significance: + p<0.10, * p<0.05, ** p<0.01.
(PDF)

**S3 Table. Regression estimates of having received first measles vaccination at age of 24 month for three groups depicted in Fig 5.** Odds-Ratios from logistic regressions using weighted and multiply imputed data; McFadden's R2 derived using Rubin's combination rules and ignoring the clustered data structure; Nimputed: due to the imputation of some missing values on the reason variables the estimation sample differ slightly between the 25 sets of imputed data; Significance: + p<0.10, * p<0.05, ** p<0.01.
(PDF)

**S4 Table. Regression estimates of vaccination behavior depicted in Fig 6.** Odds-Ratios from logistic regressions using weighted and multiply imputed data; (1) McFadden's R2 derived

using Rubin's combination rules and ignoring the clustered data structure; Significance:
+ p<0.10, * p<0.05, ** p<0.01.
(PDF)

**S1 Fig. Vaccination behavior at 24 months for all recommended vaccinations for sub-groups defined by reasons: "No reasons", "confidence reason(s)" and "complacency reason(s)".**
(TIF)

**S2 Fig. Measles vaccination rates over time by region from 1987 to 2003.** KiGGS data weighted, 14,007 observations for first measles dose by age 24 months and 9,715 observations for second dose by age 7 years, point estimates and 95 percent confidence intervals.
(TIF)

**S3 Fig. Measles vaccination rates over time by region from 2004 to 2017 based on school entry examination data.**
(TIF)

**S4 Fig. Reasons to not vaccinate over time by region.** KiGGS data weighted, depending on reason between 13,517 and 13,784 observations, point estimates and 95 percent confidence intervals.
(TIF)

**S5 Fig. Measles vaccination behavior over time for parent groups defined by (not) having deliberate (confidence, complacency) or convenience-related reasons against vaccination by region.** KiGGS data weighted, point estimates and 95 percent confidence intervals.
(TIF)

## Acknowledgments

We thank two anonymous reviewers, Boris Holzer, Sebastian Koos, Peter Preisendörfer, Britta Renner, and especially Christina Poethko-Mueller for valuable comments and suggestions on earlier versions of this paper. We are grateful to the KiGGS Study Group who provided the data and support in using it.

## Author Contributions

**Conceptualization:** Claudia Diehl.

**Data curation:** Christian Hunkler.

**Formal analysis:** Christian Hunkler.

**Writing – original draft:** Claudia Diehl.

**Writing – review & editing:** Christian Hunkler.

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
