## [Decision Letter · Decision Letter 0]

27 Oct 2021

PONE-D-21-11294

Vaccination-related attitudes and behavior across birth cohorts. Evidence from Germany

PLOS ONE

Dear Dr. Diehl,

Thank you for submitting your manuscript to PLOS ONE. After careful consideration, we feel that it has merit but does not fully meet PLOS ONE’s publication criteria as it currently stands. Therefore, we invite you to submit a revised version of the manuscript that addresses the points raised during the review process.

While I think this is an important topic that warrants investigation, there were several issues with both the study design, methodology, including research questions and statistical analyses, as well as presentation of study limitations in a separate paragraph.

Tables and figures should be checked regarding their content.

These abovementioned issues are significant enough that they seriously undermine the contributions of the study. Together with both reviewers, I have a number of reservations about this paper. The long list is presented below.

Furthermore, the writing should be proofread by a native English speaker due to the linguistic and grammatical errors.

We look forward to receiving your revised manuscript.

Kind regards,

Prof. Maria Gańczak

Academic Editor

PLOS ONE

Journal Requirements:

2. We note that you have included redacted text in your manuscript. As PLOS ONE is not a double blind peer review journal can you please insert the missing text so all details can be read in full.

Reviewers' comments:

Reviewer's Responses to Questions

**Comments to the Author**

1. Is the manuscript technically sound, and do the data support the conclusions?

Reviewer #1: Partly

Reviewer #2: Yes

2. Has the statistical analysis been performed appropriately and rigorously? 

Reviewer #1: Yes

Reviewer #2: Yes

3. Have the authors made all data underlying the findings in their manuscript fully available?

Reviewer #1: Yes

Reviewer #2: No

4. Is the manuscript presented in an intelligible fashion and written in standard English?

Reviewer #1: Yes

Reviewer #2: Yes

5. Review Comments to the Author

Reviewer #1: This paper seeks to add to the existing knowledge about trends in vaccination-related attitudes and behavior—and the link therein. It contributes to the research on the question whether vaccine hesitancy is a growing problem. This is a relevant public health question. Vaccine hesitancy is of concern to global public health and knowing whether it is an increasing problem or not contributes to understanding the phenomenon (and also its urgency). The study relies on data that has been collected 15 years ago and reports on behavior and attitudes during the 1990s. It thus gives an insight into trends in the past, thus more specifically it tries to answer the question whether attitudes and behavior have changed during an episode 20-30 years ago. The research field vaccine hesitancy has only been developing over the past 10 years, hence more advanced measures to assess vaccine hesitancy have not been available, when the data was initially collected, and have hence not been applied.

Major remarks

The authors prominently discuss the influence the internet has on trends in vaccine hesitancy. However, the reviewer is hesitant how the data collected can answer this question. The time period they are looking it is largely before the introduction of the internet. Even if the data they collected shows that independent of the internet attitudes and behavior change over time, this does not allow to judge the influence the internet has on vaccination behavior and attitudes. It might have a strong influence – and other factors might also have a strong influence. The reviewer suggests to discuss this more explicitly and to shorten the discussions about potential influence of the internet in the manuscript. The reviewer further suggests that the authors discuss more in dept, which other factors could play a role in changes in vaccination behavior and attitudes. E.g. one hypothesis they have is that health education has improved. Do they have any evidence to prove that? It would be interesting to read how the time period 1990-2000 has been in Germany with regard to health education.

The authors should include a paragraph under methods where they describe the regression analysis in detail. This should include that they used logistic regression, that the dependent variables were dichotomous (e.g. 0= not vaccinated; 1= vaccinated), that all analysis was done with Stata?, how they built the regression models, interactions, whether they looked for multicollinearity. They should further explain how they sorted the participants in different groups according to the reasons they gave for non-vaccination, e.g. an individual who gives both confidence and complacency reason, it is included in both subgroups then?

Due to the study design, the authors have only limited information on vaccine attitudes in their data. As those with deliberate reasons make a small fraction of their whole sample, the majority is individuals with “no reasons”. However, in this group with “no reasons” there are those who decide to vaccinate and those that decide against it. The items that participants could choose from (confidence, complacency) are very limited and give a limited option for the participants to indicate their motivation. The authors should discuss that those with “no reasons” probably do have reasons for non-vaccination, but that these reasons were not assessed. Moreover, it is difficult to generalize on vaccine attitudes for the whole sample from this data, e.g. a claim like the data “contains valid information on changes across cohorts in parents’ confidence in vaccination, their vaccination complacency” seems difficult to make, as they do not have data on confidence for the whole sample (p. 26 line 21). E.g. Betsch et al. developed a measure that assess confidence among all participants, thus e.g. a mean confidence for a given birth cohort could be reported.

The authors state “To be sure, one can become skeptical about vaccination later in life, but the topic should be much less salient once the decision has been made.” The reviewer is not convinced that attitudes stay as stable over time – could the authors provide more evidence here to back their methodology? Moreover, even though vaccination decisions are more frequent in early childhood, vaccination is not a topic for small children only. E.g. considering the TDaP booster, the FSME vaccine, the MenB vaccine for travelers.

The authors’ second research question is “whether or not parents with deliberate reasons against vaccination have become more likely across birth cohorts to have unvaccinated children”. How did the authors develop this hypothesis? Why did they come to focus on this research question in the first place? Was this an exploratory analysis or an hypothesis with which they approached the research project? Maybe the authors could make this clearer in the manuscript.

The reviewer suggests to include a limitations section under discussion. Limitations are given in different parts of the manuscript. The reader could benefit from finding all limitations in one place. This section should, among others, include that – as the authors note – (i) no longitudinal data was available, the method they chose is an approximation of what happens over time, (ii) comment on the way they assessed vaccine hesitancy (e.g. no validated items, items that may not capture the dimensions confidence, complacency, convenience really well, because they were used before the research field developed a clear concept of vaccine hesitancy), (iii) that this is data which captures a time period in the 1990s, thus does not allow to extrapolate a trend in the development of vaccine hesitancy in recent years, (iv) attitudes were assessed at one point in time (2003-2006) but parents should comment on their vaccination behavior a long time ago, not at the point in time where they took the decision, (v) the by far largest group was a group of individuals who did not give reasons for non-vaccination, thus for the largest groups it remains unknown why they chose not to vaccinate.

Minor remarks

The reviewer suggests to shorten the introduction and place the hypothesis/ research questions in one place for better readability of the manuscript. To the reviewer it was difficult to grasp the research questions as they seemed to occur twice in the introduction and seemed to vary.

The authors mention institutional trust or mistrust and its relationship with vaccine hesitancy in the introduction. However, it is unclear how this relates to their findings, as they do not assess institutional trust in their study.

The authors links their results to COVID-19 and the current interest in vaccine hesitancy in the verge of the pandemic. The review suggests to shorten this part, as it does not lead the reader well to measles vaccination behavior and attitudes 20-30 years ago.

The authors should mention the surveys on vaccination attitudes and behaviors conducted by the Federal Centre for Health Education. This is cross-sectional data on vaccination behavior and attitudes during the last 5-10 years. https://www.bzga.de/forschung/studien/abgeschlossene-studien/studien-ab-1997/impfen-und-hygiene/

p. 4 line 7. The authors should replace this reference with a more recent reference, covering vaccination coverage at school entry 2008-2018. Rieck et al. 2020 https://edoc.rki.de/bitstream/handle/176904/6902.4/32-33_2020_DOI_Impfquoten_Version%203.pdf?sequence=9&isAllowed=y

The authors might want to change the threshold for herd immunity (60-70 percent for Covid-19). To the reviewer’s knowledge no such threshold is currently known and if there was one it is estimated to be higher than 60-70 percent.

The authors might want to change the terminology from vaccination hesitancy to vaccine hesitancy.

p. 9 line 11. The authors might want to consider a more recent publication on physician recommendation behavior: Neufeind et al. 2020. https://www.sciencedirect.com/science/article/pii/S0264410X20305466

The survey was conducted between May 2003 and May 2006. The reviewer would like to know the authors thoughts about whether this is methodologically a problem, as attitudes might have changed between 2003 and 2006? It the authors think so they might want to consider to add this under limitations.

p.12 line 15. The authors write that they “shifted the upper horizontal axis five years to the left to roughly “match” with the rates calculated from the KiGGS data that are displayed by birth cohort”. The reviewer has difficulty understanding what the authors did here and why. Maybe they could add further explanation for better understanding? How do the authors account for the difference in the age groups between KiGGS (24 months and 7 years) and school entry exam (4-6 years)?

The authors generally in the figures report the sample size of e.g. those with no reasons, they do not give the sample size for each birth cohort. The is probably to keep it simple, however thus the reader has a skewed understanding of the confidence intervals. Maybe the authors could think of a way to give the reader an understanding of the sample size in the respective birth cohorts?

The group with the category lowest education seems to be very small. Is this groups underrepresented in the sample? If so, how did the authors address this problem? Was this a problem in their regression models?

Could the authors explain why time intervals chosen varied? The first time intervals seem to be four years long, the last one eight years? Does it make a difference in the results when those intervals are changed?

The manuscript might benefit from a language check and could be written in a more concise way.

Reviewer #2: This is an interesting and well-written article. Some questions/concerns that would be good to address in the final version:

From what I understand, parents were asked the reasons for not vaccinating their children. The same question was asked of parents of young and older children. No comment is made about the potential recall bias associated with parents of older children being asked to comment on reasons for not vaccinating when those decisions were many years ago. How has this been accounted for in the analysis?

The purpose of using the entry exam data is not clear and this data is not fully described or presented.

The number of people providing different responses is not clear. Table 1 should have numbers also, not just %.

Fig 5 is missing a full label for the y axis. % of what?

Sentence: "Living in a migrant family reduces the chances of being vaccinated for children whose parents do not report any reasons against vaccination". -- Compared to which group? The following sentence is also not clear.

6. PLOS authors have the option to publish the peer review history of their article (what does this mean?). If published, this will include your full peer review and any attached files.

Reviewer #1: No

Reviewer #2: No

---

## [Author Response · Author response to Decision Letter 0]

13 Dec 2021

please see attched response-to-reviewers file

---

## [Decision Letter · Decision Letter 1]

31 Jan 2022

Vaccination-related attitudes and behavior across birth cohorts. Evidence from Germany

PONE-D-21-11294R1

Dear Dr. Diehl%,

We’re pleased to inform you that your manuscript has been judged scientifically suitable for publication and will be formally accepted for publication once it meets all outstanding technical requirements.

Kind regards,

Prof. Maria Gańczak

Section Editor

PLOS ONE

Additional Editor Comments (optional):

Reviewers' comments:

Reviewer's Responses to Questions

**Comments to the Author**

1. If the authors have adequately addressed your comments raised in a previous round of review and you feel that this manuscript is now acceptable for publication, you may indicate that here to bypass the “Comments to the Author” section, enter your conflict of interest statement in the “Confidential to Editor” section, and submit your "Accept" recommendation.

Reviewer #1: All comments have been addressed

Reviewer #2: All comments have been addressed

2. Is the manuscript technically sound, and do the data support the conclusions?

Reviewer #1: Yes

Reviewer #2: Yes

3. Has the statistical analysis been performed appropriately and rigorously? 

Reviewer #1: I Don't Know

Reviewer #2: Yes

4. Have the authors made all data underlying the findings in their manuscript fully available?

Reviewer #1: Yes

Reviewer #2: Yes

5. Is the manuscript presented in an intelligible fashion and written in standard English?

Reviewer #1: Yes

Reviewer #2: Yes

6. Review Comments to the Author

Reviewer #1: Dear colleagues,

The authors have revised the manuscript and thereby addressed all the points the reviewer has raised in the first review. The changes made are comprehensive, intelligible and fully satisfactory.

The reviewer has no further comments and wishes the authors all the best for their paper.

With regard to statistical analysis the reviewer has no expertise in imputation.

Reviewer #2: (No Response)

7. PLOS authors have the option to publish the peer review history of their article (what does this mean?). If published, this will include your full peer review and any attached files.

Reviewer #1: No

Reviewer #2: **Yes: **Jody Tate

---

## [Editor Report · Acceptance letter]

3 Feb 2022

PONE-D-21-11294R1 

Vaccination-related attitudes and behavior across birth cohorts:
Evidence from Germany 

Dear Dr. Diehl:

I'm pleased to inform you that your manuscript has been deemed suitable for publication in PLOS ONE. Congratulations! Your manuscript is now with our production department. 

Kind regards, 

on behalf of

Prof. Maria Gańczak 

Section Editor

PLOS ONE